# Reversing chemorefraction in colorectal cancer cells by controlling mucin secretion

**Gerard Cantero-Recasens[1]\*[†], Josune Alonso-Marañón[2†], Teresa Lobo-Jarne[2], Marta Garrido[2], Mar Iglesias[3], Lluis Espinosa[2]\*[‡], Vivek Malhotra[4,5]\*[‡]**

[1]Renal Physiopathology Group, Vall d'Hebron Research Institute (VHIR), Barcelona, Spain; [2]Cancer Research Program, Institut Mar d'Investigacions Mèdiques, CIBERONC Hospital del Mar, Barcelona, Spain; [3]Department of Pathology, Institut Mar d'Investigacions Mèdiques, Universitat Autònoma de Barcelona, CIBERONC, Barcelona, Spain; [4]Centre for Genomic Regulation (CRG), The Barcelona Institute for Science and Technology, Barcelona, Spain; [5]Institució Catalana de Recerca i Estudis Avançats (ICREA), Barcelona, Spain

**\*For correspondence:**
gerard.cantero@vhir.org (GC-R);
lespinosa@imim.es (LE);
vivek.malhotra@crg.eu (VM)

[†]These authors contributed equally to this work
[‡]These authors also contributed equally to this work

**Abstract** Fifteen percent of colorectal cancer (CRC) cells exhibit a mucin hypersecretory phenotype, which is suggested to provide resistance to immune surveillance and chemotherapy. We now formally show that CRC cells build a barrier to chemotherapeutics by increasing mucins' secretion. We show that low levels of KChIP3, a negative regulator of mucin secretion (Cantero-Recasens et al., 2018), is a risk factor for CRC patients' relapse in a subset of untreated tumours. Our results also reveal that cells depleted of KChIP3 are four times more resistant (measured as cell viability and DNA damage) to chemotherapeutics 5-fluorouracil + irinotecan (5-FU+iri.) compared to control cells, whereas KChIP3-overexpressing cells are 10 times more sensitive to killing by chemotherapeutics. A similar increase in tumour cell death is observed upon chemical inhibition of mucin secretion by the sodium/calcium exchanger (NCX) blockers (Mitrovic et al., 2013). Finally, sensitivity of CRC patient-derived organoids to 5-FU+iri. increases 40-fold upon mucin secretion inhibition. Reducing mucin secretion thus provides a means to control chemoresistance of mucinous CRC cells and other mucinous tumours.

## Editor's evaluation

Hypersecretion of mucins by colorectal cancer (CRC) cells confers resistance to immune surveillance, and this phenomenon has also been postulated to confer resistance to chemotherapy. This work describes studies aimed at investigating mucus secretion and proteins influencing mucus secretion and the impact on the efficacy of a commonly used chemotherapy treatment. These results shed light on the chemoresistance of mucinous tumors and point to a possible prognostic marker involved in mucus secretion.

## Introduction

Secreted mucins are synthesized in the endoplasmic reticulum (ER) and transported to the Golgi apparatus, where they receive extensive O-glycosylation that increases their molecular weight by about fivefold. These heavily glycosylated mucins are then packed into specialized micrometre-sized granules where they can reach molecular weights of up to 50 MDa (*Sheehan et al., 2004*; *Thornton et al., 2008*). These granules mature by a process that involves mucin condensation, and, finally, a small

number of granules fuse to plasma membrane in a calcium-dependent manner to release the required quantity of mucins in the extracellular space (*Adler et al., 2013*; *Rossi et al., 2004*). Specifically, mucins are secreted by two main processes: (1) extracellular calcium-dependent pathway called stimulated secretion, which is triggered by an exogenous agonist; and (2) intracellular calcium-dependent release, but independent of external agonists, called baseline secretion. It has recently been shown that cooperation between TRPM4/5 sodium channels and the $Na^+/Ca^{2+}$ exchangers (NCX) controls stimulated secretion (*Mitrovic et al., 2013*; *Cantero-Recasens et al., 2019*). The calcium-binding protein KChIP3, on the other hand, regulates baseline secretion (*Cantero-Recasens et al., 2018*).

The secreted mucins compose the mucus that protects the underlying epithelium from toxins, pathogens, and allergens (*Kesimer et al., 2013*). Secretion of the right quantity and quality of mucins is crucial because mucin hypersecretion in the airways is linked to chronic obstructive diseases and hyposecretion makes the underlying tissue more susceptible to damage by the immune cells and the foreign particles (*Thornton et al., 2008*; *Pelaseyed et al., 2014*). Most cancer cells start expressing mucins, and it is widely thought that they isolate tumour cells from the immune system and also render them resistant to anticancer therapies (*Kufe, 2009*). Indeed, treatment with O-glycosylation inhibitors (benzyl-α-GalNAc) increases the effectiveness of chemotherapy on pancreatic cells overexpressing the transmembrane mucin MUC1 (*Kalra and Campbell, 2007*). But whether this is due to inhibition of MUC1 glycosylation only or any of the other O-glycosylated proteins is not known. Patients with mucinous colorectal adenocarcinoma (10–15% of colorectal cancer [CRC]; *Luo et al., 2019*), however, present dysregulated mucin secretion that is long associated with high resistance to chemotherapy. In fact, extracellular mucins account for at least 50% of the volume of these tumours (*Hugen et al., 2016*). Whether or not controlling the levels of mucins secreted by cancer cells affects their susceptibility to chemotherapeutics, however, remains untested. Here, we have studied this question directly in CRC cell lines and organoids derived from a CRC patient and show that reducing mucin secretion reverses the chemorefractory properties of hypersecretory CRC cells.

## Results

### 5-Fluorouracil + irinotecan treatment induces mucin production

15–20% of CRCs develop a mucinous profile (*Luo et al., 2019*), which renders cancer cells highly resistant to commonly used chemotherapies, such as 5-fluorouracil + irinotecan (5-FU+ iri.) (*Van Cutsem et al., 2016*). To study the contribution of secreted mucins to CRC chemorefraction, we used the HT29 CRC cell line (clones HT29-18N2 and HT29-M6) that are differentiated into mucin-producing cells (*Mitrovic et al., 2013*; *Mayo et al., 2007*).

We first tested the effect of 5-FU + iri. on mucin production, focusing on secreted mucins, which are the main macrocomponents of the mucus layer (i.e. MUC5AC) (*Thornton et al., 2008*). Briefly, differentiated HT29-M6 cells were treated with 5-FU + iri. (50 µg/mL 5-fluorouracil + 20 µg/mL irinotecan diluted in normal tissue culture medium) and MUC5AC levels monitored over time (0, 1, 3, 6, and 24 hr) by Western blot (WB) (scheme in *Figure 1—figure supplement 1A*). Our results reveal that 5-FU + iri. treatment strongly increases MUC5AC intracellular levels (*Figure 1A*). To further confirm this effect, 5-FU + iri.-treated HT29-M6 cells (24 hr) were imaged by confocal immunofluorescence microscopy. Accordingly, we found that 5-FU + iri.-treated cells present a clear increase in MUC5AC level compared to control cells (*Figure 1B*). These results suggest that increased production of mucins after 5-FU + iri. treatment could act as a barrier to chemical permeation, thus enhancing chemorefraction of mucosecretory cancer subtypes.

Differentiated HT29-M6 cells were seeded in glass-bottom dishes and treated with 100 µM ATP, a well-known secretagogue for mucins (*Abdullah et al., 1997*). After 30 min, cells were divided into two groups: one group was extensively washed with isotonic solution to remove the mucin fibres, while the other group was subjected to mild washes to preserve these fibres. Next, cells were incubated with Alexa Fluor 488-labelled albumin for 1 hr, processed for confocal immunofluorescence microscopy and stained with anti-MUC5AC antibody. As shown in *Figure 1C*, mucins act as a barrier that prevents the endocytosis of albumin by HT29-M6 cells (*Figure 1C*, upper panel). Contrarily, removal of mucin fibres allows the contact and uptake of albumin by cells (*Figure 1C*, lower panel). Quantification of the images using the Fiji software revealed that in the presence of mucin fibres albumin colocalizes with MUC5AC (Manders' coefficient albumin-MUC5AC: 0.16 vs. 0.01, p=0.01), whereas in cells without

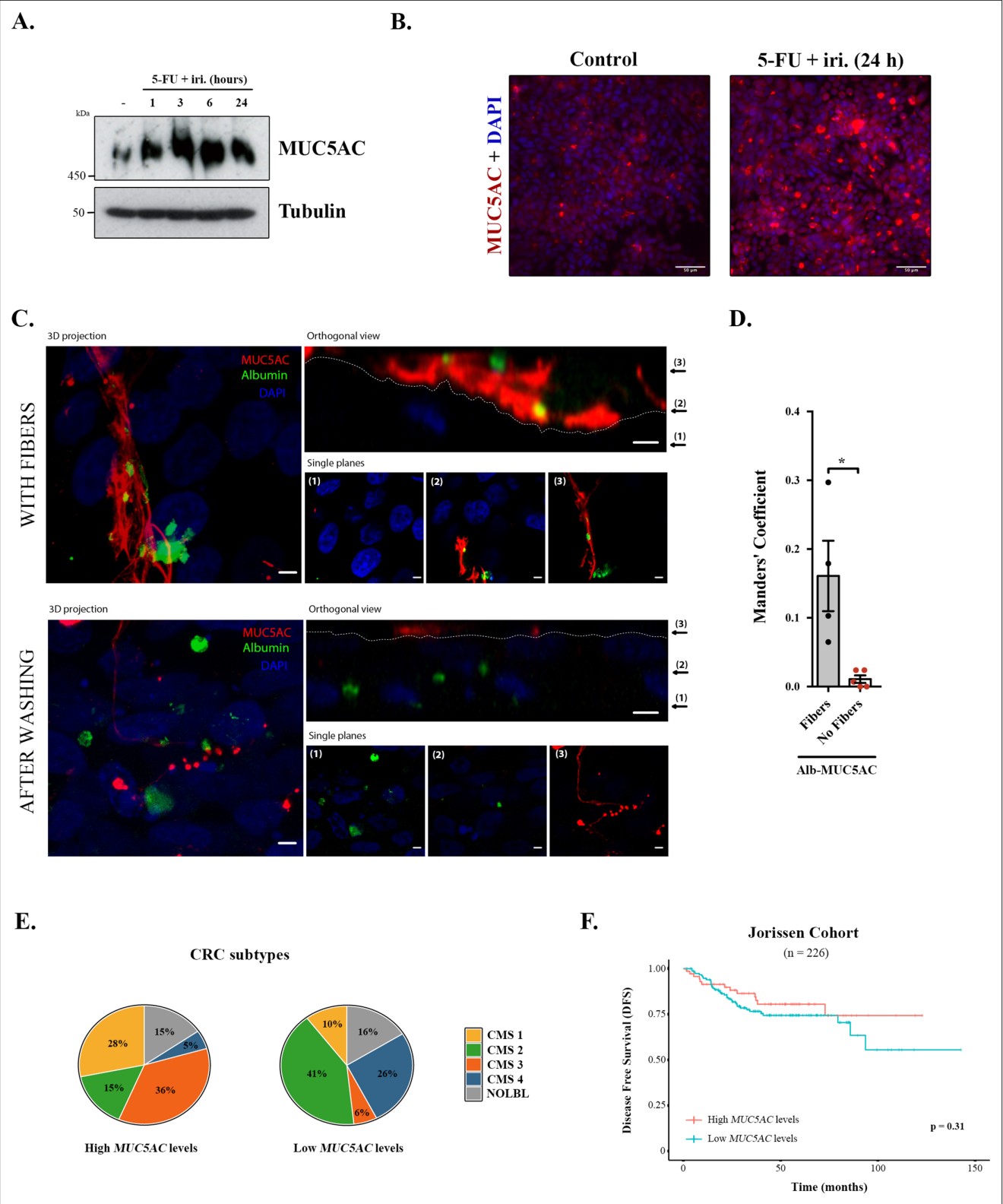

**Figure 1.** Mucins in colorectal cancer. (**A**) Cell lysates from differentiated HT29-M6 cells treated with 5-fluorouracil + irinotecan (5-FU + iri.) (50 μg/mL 5-FU + 20 μg/mL iri.) for 24 hr were analysed by Western blot with an anti-MUC5AC to test expression levels. Tubulin was used as a loading control. (**B**) Immunofluorescence Z-stack projections of differentiated HT29-M6 cells treated with vehicle (control) or 5-FU + iri. (50 μg/mL 5-FU + 20 μg/mL irinotecan) for 24 hr. Cells were stained with anti-MUC5AC (red) and DAPI (blue). Scale bar = 50 μm. (**C**) Differentiated HT29 cells were treated with 100 μM ATP for 30 min, washed extensively (upper panel) or smoothly (lower panel), and incubated with Alexa Fluor 488-labelled albumin for 1 hr. Cells

*Figure 1 continued on next page*

*Figure 1 continued*

were stained with anti-MUC5AC antibody (red) and DAPI (blue). In the orthogonal view, dotted lines across the images demarcate the top surface of the cell. Scale bars correspond to 5 μm. (**D**) Colocalization between MUC5AC and albumin was calculated from immunofluorescence images by Manders' coefficient using Fiji. Average values ± SEM are plotted as scatter plot with bar graph. The y-axis represents Manders' coefficient of the fraction of albumin overlapping with MUC5AC (N ≥ 3). (**E**) Consensus molecular subtype (CMS) distribution of high and low MUC5AC-expressing colorectal tumours (Jorissen cohort). (**F**) Disease-free survival (DFS) of colorectal cancer patients with high (n = 73) or low (n = 153) MUC5AC levels (Jorissen cohort). NOLBL: samples without a defined CMS subtype (tumours with no label). *p<0.05, **p<0.01.

The online version of this article includes the following source data and figure supplement(s) for figure 1:

**Source data 1.** Uncropped gels for *Figure 1*.

**Figure supplement 1.** Experimental design for measuring mucin production after treatment.

mucin barrier, albumin is found intracellularly (*Figure 1D*). Altogether, these data demonstrate that colonic cancer cells produce higher levels of MUC5AC in response to chemotherapy treatment, which form a barrier that protects cells from incorporation of extracellular compounds. We anticipated that MUC5AC may act as a barrier against chemotherapy in human CRC tumours, and as a consequence, it could represent a prognostic biomarker for CRC.

Analysis of CRC subtypes, using the consensus molecular subgroup (CMS) classification from *Guinney et al., 2015*, indicated a radically different distribution of high and low MUC5AC-expressing tumours among subtypes: CMS1 (28% vs. 10%), CMS2 (15% vs. 41%), CMS3 (36% vs. 6%), and CMS4 (5% vs. 26%) (*Figure 1E*). Importantly, both CMS1 (associated with immune evasion) and CMS3 (metabolic dysregulation and goblet cell markers enrichment) have worse prognosis than CMS2 (*Guinney et al., 2015*), which include the majority of tumours with low MUC5AC expression. These results point to an important role of MUC5AC in the prognosis of CRC patients. To directly assess this possibility, we evaluated whether MUC5AC expression levels discriminate patients' outcome in a well-annotated CRC cohort (GSE14333, *Jorissen et al., 2009*). Importantly, although this public database does not include information on chemotherapy treatment, we could analyse the disease-free survival (DFS) of patients depending on the gene-expression levels, a parameter that is related to tumour chemoresistance. However, DFS analysis revealed no significant differences between patients with high or low MUC5AC levels (*Figure 1F*), strongly suggesting that factors controlling mucin secretion, and not only intracellular mucin levels, might be relevant biomarkers for CRC prognosis.

## The mucin regulator KChIP3 is a risk factor for CRC with high MUC5AC expression

Mucin secretion can be triggered by an external agonist (stimulated secretion), such as ATP and allergens, or in the absence of external agonists (baseline secretion) (*Adler et al., 2013*). We have shown previously that KChIP3, an intracellular calcium sensor, negatively controls baseline mucin secretion (loss of KChIP3 causes mucin hypersecretion) (*Cantero-Recasens et al., 2018*). So, we checked whether KChIP3 could be a prognostic biomarker for CRC. Analysis of DFS of patients according to KChIP3 levels, encoded by the *KCNIP3* gene, revealed that KChIP3 is, indeed, a prognostic factor (hazard ratio [HR] = 1.925, p=0.02) (*Figure 2A*). To further confirm whether this effect is related to mucins, patients were divided between high (n = 73) and low (n = 153) MUC5AC expression levels, and then we studied the effect of KChIP3 on the DFS of patients. Our analysis reveals that low levels of KChIP3 are a clear risk factor in those patients with high MUC5AC levels (HR = 3.29, p=0.04) (*Figure 2B*). There is no significant effect on patients with low MUC5AC expression (*Figure 2C*), although after 100 months there is a decrease in DFS. Further analysis of this effect revealed that three patients (GSM358532, GSM358534, and GSM358438) were responsible for this effect. We studied these patients in more detail, and interestingly, although they have low levels of MUC5AC, these patients present increased levels of other secreted mucins (e.g., MUC2) (*Figure 2—figure supplement 1A*). However, similarly to previous studies (*Kufe, 2009*; *Luo et al., 2019*), analysis of the effect of MUC2 levels on DFS shows that high MUC2 levels are protective while patients with low levels of MUC2, which may reflect a more dedifferentiated state of cancer cells, have a lower DFS (HR = 2.54, p=0.023) (*Figure 2—figure supplement 1B*). Importantly, when we studied the effect of KChIP3 on the DFS of patients with high expression of MUC2, we found a clear tendency (p=0.08) to worse prognosis in patients with low levels of KChIP3 (*Figure 2—figure supplement 1C*). Altogether, these results suggest that low levels

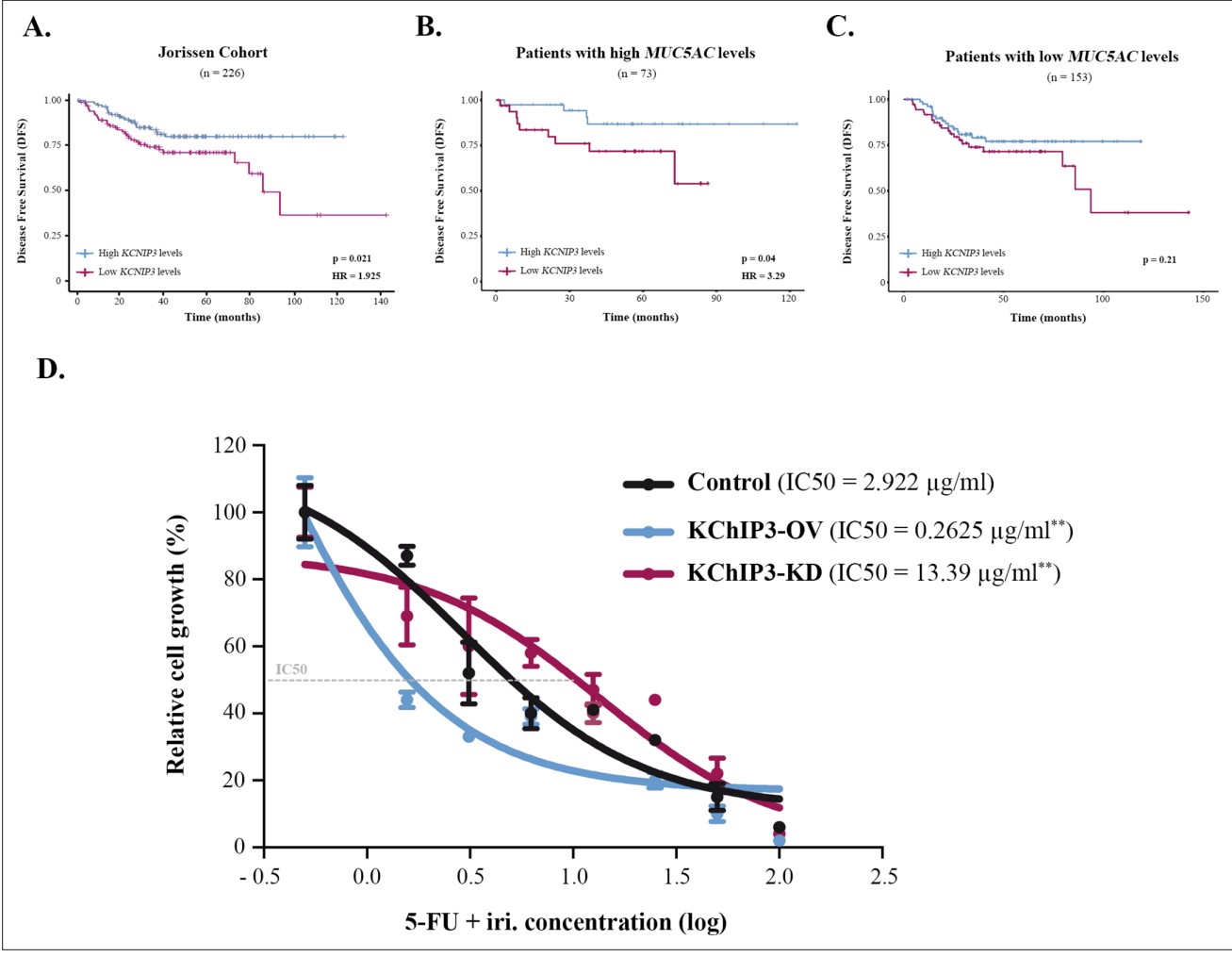

**Figure 2.** KChIP3 is a prognostic marker of colorectal cancer (CRC). (**A–C**) Disease-free survival (DFS) according to KChIP3 levels of CRC patients (low KChIP3 levels, n = 120; high KChIP3 levels, n = 106) (**A**), CRC patients with high MUC5AC levels (low KChIP3 levels, n = 39; high KChIP3 levels, n = 34) (**B**), or CRC with low MUC5AC levels (low KChIP3 levels, n = 81; high KChIP3 levels, n = 72) (**C**). (**D**) Differentiated control (black), KChIP3-overexpressing cells (KCNIP3-OV, blue) and KChIP3-depleted cells (KCNIP3-KD, red) were treated for 72 hr with increasing concentrations of 5-fluorouracil + irinotecan (5-FU + iri.). Average values ± SEM are plotted as scatter plot (N > 3). The y-axis represents the percentage of cell growth relative to the lowest concentration of 5-FU + iri. The IC50 was calculated from the interpolated curve. HR, hazard ratio. *p<0.05, **p<0.01.

The online version of this article includes the following figure supplement(s) for figure 2:

**Figure supplement 1.** Expression of mucins in patients.

of KChIP3, which consequently increase the quantity of mucins secreted, protect cancer cells from chemotherapeutic drugs.

## Modulation of KChIP3 levels alters CRC cells' chemoresistance

To further test this hypothesis and the role of KChIP3 in chemoresistance, we used HT29-18N2 cell lines stably depleted or overexpressing KChIP3 (KChIP3-KD or KChIP3-OV, respectively) (***Cantero-Recasens et al., 2018***). Briefly, differentiated control, KChIP3-KD (increased mucin secretion), and KChIP3-OV (reduced mucin release) cells were treated with increasing concentrations of 5-FU + iri. and then monitored for cell viability using the CellTiter-Glo assay. As shown in ***Figure 2D***, cells depleted of KChIP3 are fourfold more resistant to 5-FU + iri. than control cells, whereas overexpression of KChIP3 renders cancer cells almost 10 times more sensitive to the chemotherapy (IC50 control cells = 2.9 µg/mL, KChIP3-KD cells = 13.4 µg/mL, KChIP3-OV = 0.3 µg/mL) (***Figure 2D***).

In summary, low levels of KChIP3 increase mucin secretion (*Cantero-Recasens et al., 2018*), which protects CRC cells from chemotherapy. Contrarily, high levels of KChIP3 inhibit baseline mucin release, thereby reducing the mucus barrier and rendering cancer cells more sensitive to 5-FU+ iri.

## Inhibition of MUC5AC secretion enhances the effect of chemotherapy

Although baseline mucin secretion and its regulator KChIP3 have a clear role in the response to chemotherapy of CRC cells, our results indicated that 5-FU + iri. treatment can trigger mucin production. Interestingly, it was previously described that stimulated mucin secretion depends on extracellular calcium entry through TRPM4/5 – $Na^+/Ca^{2+}$ exchangers (NCX) cooperation (*Cantero-Recasens et al., 2019*). We anticipated that if CT-induced mucin secretion was increasing CRC chemoresistance, benzamil, an NCX inhibitor (*Watanabe et al., 2006*), should enhance the effect of 5-FU + iri. on HT29-M6 CRC cells. Differentiated HT29-M6 cells were treated with 5-FU + iri. (IC50) and 20 μM benzamil or vehicle, and the levels of DNA damage were determined by the standard comet assay (*Colomer et al., 2019*). Our results showed that co-treatment with 5-FU + iri. and benzamil increased the levels of DNA breaks compared to 5-FU + iri. plus vehicle (*Figure 3A*). We then tested cell viability of HT29-M6 at 24 and 72 hr post-treatment of 5-FU + iri. alone or in combination with benzamil. There was no significant effect of the combined benzamil and 5-FU + iri. treatment at 24 hr. At 72 hr, however, the therapeutic effect of low concentration of 5-FU + iri. (15% reduction on cell viability) was strongly enhanced when combined with 20 μM benzamil (40% reduction on cell viability compared with the control) (*Figure 3B*). These results demonstrate that treatment with the NCX inhibitor benzamil improves the effect of chemotherapy on tumour cell eradication.

To further confirm that this additive effect was due to the inhibition of 5-FU + iri.-dependent mucin secretion, we used a second inhibitor called SN-6, which is a specific blocker of the reverse mode (calcium influx) of NCX and can be used at lower concentration, therefore presenting fewer side effects than benzamil (*Niu et al., 2007*). HT29-M6 cells were grown at post-confluence (differentiation) in glass-bottom dishes and then left untreated or treated with 5-FU + iri. (IC50) in combination with vehicle, 20 μM benzamil, or 10 μM SN-6. Next, cells were processed for immunofluorescence with anti-MUC5AC antibody and imaged by confocal microscopy to visualize mucin granules (*Figure 3C*). Analyses of images using Fiji software (*Schindelin et al., 2012*) revealed that cells treated with 5-FU + iri. contained significantly fewer MUC5AC-containing particles than control cells (75.7 vs. 100.7 particles/field, p=0.002) with a reduced average size (1.12 μm$^3$ vs. 2.48 μm$^3$) (*Figure 3C*, quantification in *Figure 3D*). It is important to note that this anti-MUC5AC antibody mainly recognizes the mature mucin granules (close to the plasma membrane, as shown in *Cantero-Recasens et al., 2018*), which can explain the apparent contradiction with the effect of 5-FU + iri. treatment on mucin production (*Figure 1*). Besides, it has been previously shown that after maximum stimulated secretion mucin stores require 24 hr to be fully restored (*Zhu et al., 2015*). Thus, this decrease in the numbers (and size) of mucin granules is in accordance with enhanced mucin secretion by 5-FU + iri. treatment as shown in *Figure 4*. Importantly, co-treatment with benzamil or SN-6 reverted this effect, causing a strong accumulation of mucin granules, as shown by an increase in the number of MUC5AC-containing particles (benzamil: 1.88 particles/field, SN-6: 2.19 particles/field) and their size (benzamil: 134.3 μm$^3$, SN-6: 120.5 μm$^3$), compared to cells treated with 5-FU + iri. and vehicle (*Figure 3C*, quantification in *Figure 3D*). Interestingly, SN-6 had a stronger effect than benzamil on cells treated with 5-FU + iri., further indicating that NCX inhibition enhanced sensitivity to chemotherapy.

## Specific inhibition of NCX enhances 5-FU + iri.-dependent damage in both cells and patient-derived organoids

In order to test whether SN-6 effect observed on mucins correlated with increased sensitivity to 5-FU + iri. treatment, we determined the levels of γH2AX, a standard marker of genotoxic damage (*Podhorecka et al., 2010*) in HT29-M6 cells treated with 5-FU + iri. (IC50) in the presence or absence of 10 μM SN-6 (diluted in normal tissue culture medium). We obtained cell lysates at different time points (0, 1, 3, and 6 hr) after treatment and then measured the levels of MUC5AC and γH2AX by WB. Our results confirmed that SN-6 leads to MUC5AC intracellular accumulation over time in 5-FU + iri.-treated cells (*Figure 4A*, first lane), which correlated with a slight increase in γH2AX levels (*Figure 4A*, second lane). MUC5AC secretion was also tested in these cells by collecting the extracellular medium at the same time points. Importantly, the first time point (time 0) represents HT29 cells not treated

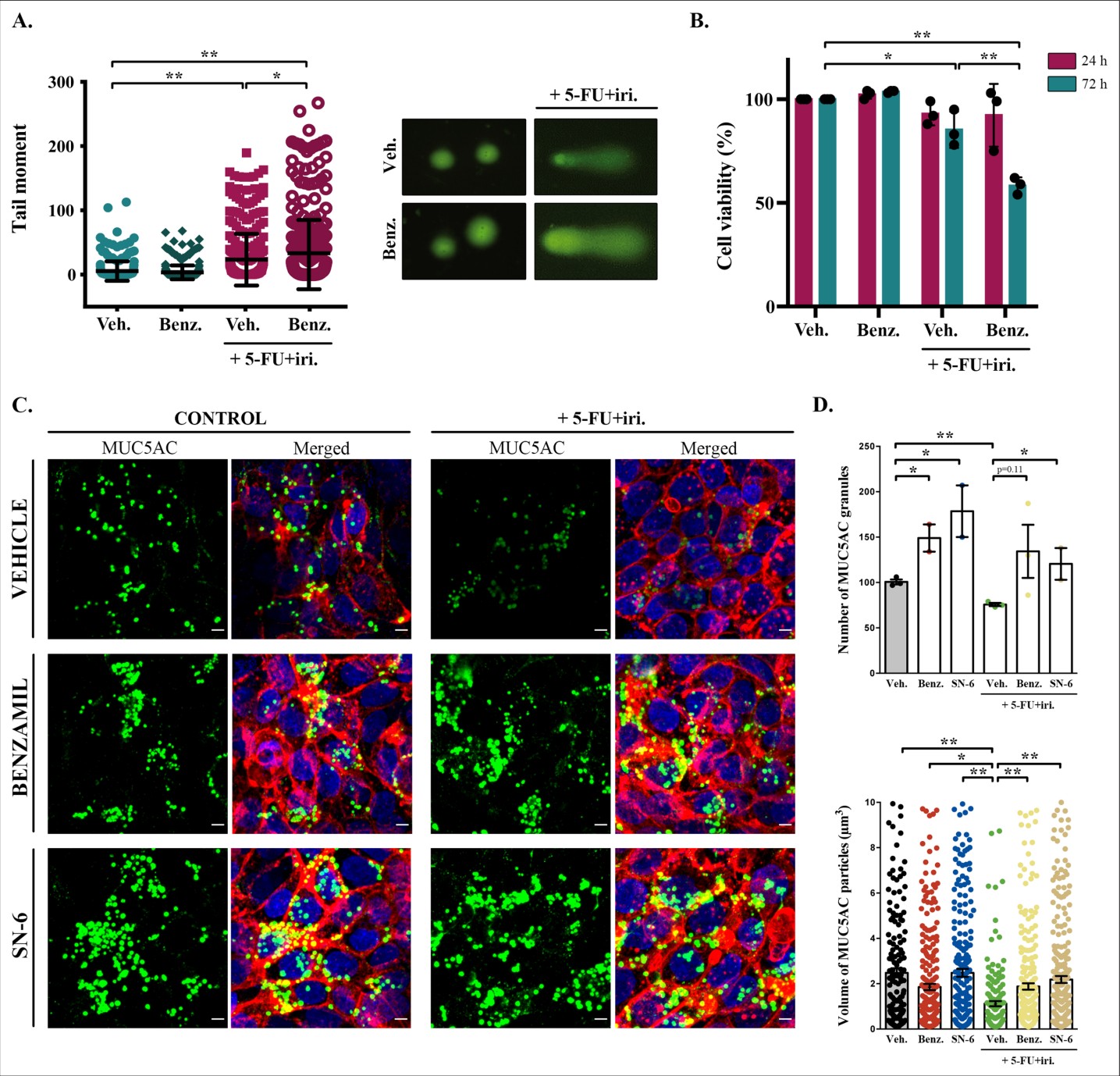

**Figure 3.** Inhibition of sodium/calcium exchangers (NCXs) enhances cell death by 5-fluorouracil + irinotecan (5-FU+ iri.). (**A**) Comet assay of HT29-M6 cell line treated with 5-FU + iri. (25 µg/mL 5-FU + 10 µg/mL irinotecan) and 20 µM benzamil, alone or in combination. The tail moment was measured after 72 hr of treatment (N > 3). (**B**) Quantification of cell viability in HT29-M6 cell line after treatment (24 or 72 hr) with 5-FU + iri. (10 µg/mL 5-FU + 4 µg/mL Irinotecan) and 20 µM benzamil, alone or in combination (N ≥ 3). (**C**) Immunofluorescence Z-stack projections of differentiated HT29-M6 cells treated with vehicle, 20 µM benzamil, or 10 µM SN-6 in the presence or absence of 5-FU + iri. Cells were stained with anti-MUC5AC (green), phalloidin (red), and DAPI (blue). Scale bar = 5 µm. (**D**) Quantification of the number (upper graph) and volume (lower graph) of MUC5AC granules from immunofluorescence images by confocal microscope. Average values ± SEM are plotted as scatter plot with bar graph (N ≥ 3). Veh., vehicle; Benz., benzamil. *p<0.05, **p<0.01.

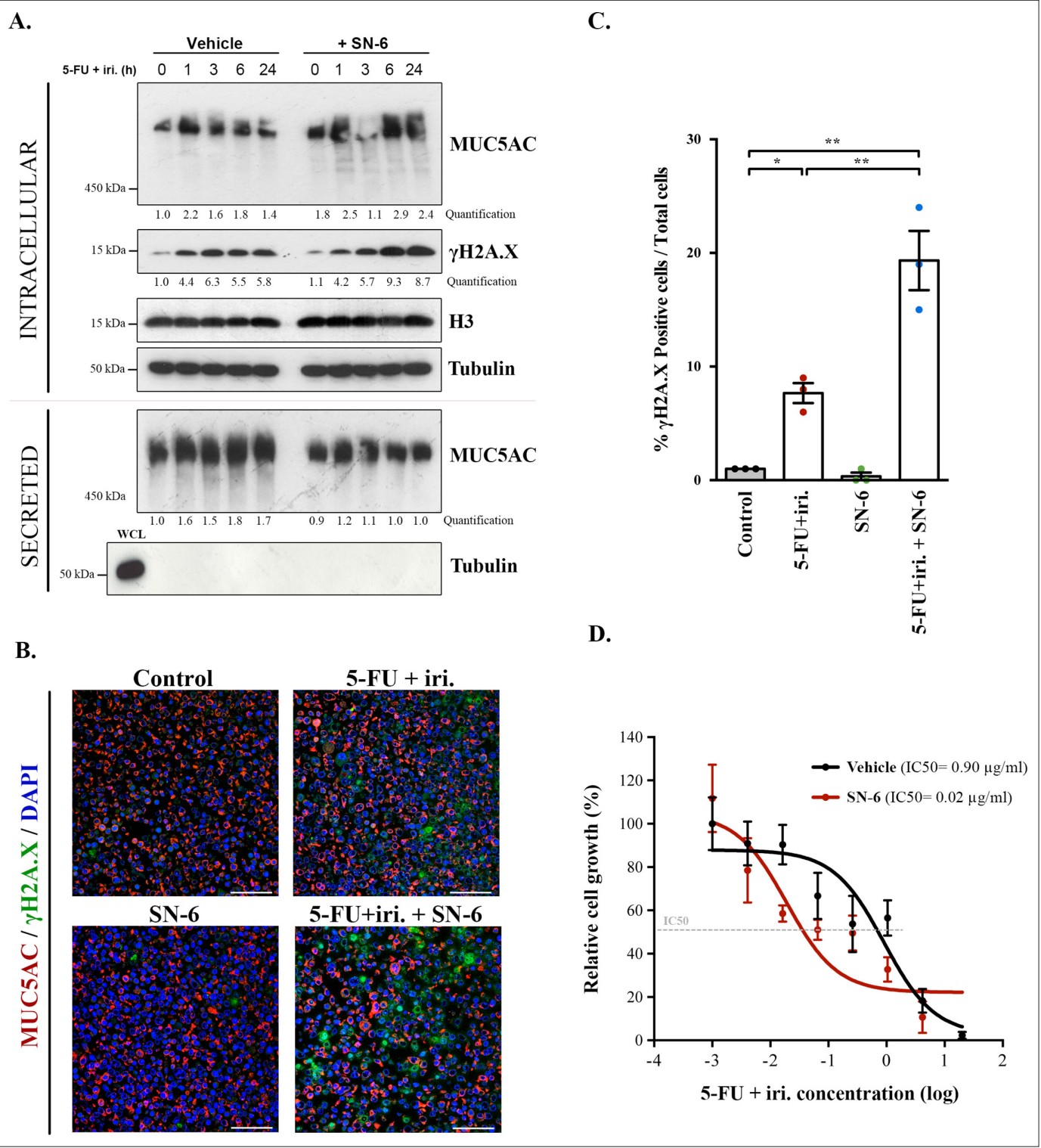

**Figure 4.** SN-6 treatment increases sensitivity of colorectal cancer (CRC)-derived cells and organoids to 5-fluorouracil + irinotecan (5-FU + iri.). (**A**) Cell lysates and secreted medium of differentiated HT29-M6 cells pre-treated with a vehicle or 10 µM SN-6 inhibitor for 24 hr and then exposed to 5-FU + iri. (50 µg/mL 5-FU + 20 µg/mL iri.) for 0, 1, 3, 6, and 24 hr were analysed by Western blot with an anti-MUC5AC, anti-γH2A.X, and H3 to test their levels. Tubulin was used as a loading control. Quantification of MUC5AC and γH2A.X is included (N ≥ 3). (**B**) Immunofluorescence images of differentiated HT29-M6 cells treated with vehicle (control) or 10 µM SN-6 in the presence or absence of 5-FU + iri. (50 µg/mL 5-FU + 20 µg/mL irinotecan) for 24 hr. Cells were stained with anti-MUC5AC (red), anti-γH2A.X (green), and DAPI (blue). Scale bar = 50 µm. (**C**) Quantification of the number of γH2A.X-positive cells in the different conditions relative to the total number of cells from the immunofluorescence images (N ≥ 3). (**D**) CRC patient-derived

*Figure 4 continued on next page*

*Figure 4 continued*

organoids (PDOs) were treated for 72 hr with increasing concentrations of 5-FU + iri. with vehicle or 10 µM SN-6. Average values ± SEM are plotted as scatter plot. The y-axis represents the percentage of cell growth relative to the lowest concentration of 5-FU + iri. The IC50 was calculated from the interpolated curve (N ≥ 3). *p<0.05, **p<0.01.

The online version of this article includes the following source data and figure supplement(s) for figure 4:

**Source data 1.** Uncropped gels for *Figure 4*.

**Figure supplement 1.** Mucins' levels in HT29 cells and in a patient-derived organoid.

with 5-FU + iri. or mucin secretion inhibitors (scheme in *Figure 4—figure supplement 1A*). Thus, our results revealed that 5-FU + iri treatment caused an increase in mucin secretion (60–80% compared to time 0), which was blocked by SN-6 (relative levels: 1.8 vs. 1.0, respectively) (*Figure 4A*). Next, to validate the increase in DNA damage, differentiated HT29-M6 cells were treated with vehicle (control), 5-FU + iri. (IC50), 10 µM SN-6 or 5-FU + iri.+ SN-6, and processed for immunofluorescence with anti-MUC5AC and anti-γH2AX antibodies (*Figure 4B*). Quantification of γH2AX-positive cells confirmed the increase on DNA damage in cells treated with 5-FU + iri. and SN-6 (control: 0.9%; SN-6: 0.4%; 5-FU + iri.: 8.0%; 5-FU + iri.+ SN-6: 24.8%) (*Figure 4C*). Mucin production was further confirmed by Alcian blue staining (*Figure 4—figure supplement 1B*). In addition, to test for the transcriptional effect of SN-6 on 5-FU + iri.-dependent mucin production, we analysed RNA levels of MUC5AC in HT29-M6 cells treated with vehicle (control), 5-FU + iri. (IC50), 10 µM SN-6, or 5-FU + iri. + SN-6. Our results revealed that the increase in mucin production and secretion by 5-FU + iri. and the effect of SN-6 on 5-FU + iri. are independent of MUC5AC transcription (*Figure 4—figure supplement 1C*).

Finally, to test the pathophysiological relevance of our findings, we studied the effect of SN-6 + 5-FU + iri. combination on a CRC patient-derived organoid (PDO). With this aim, we treated PDO cells, which we recently validated as model for therapy prescription (*Colomer et al., 2019*), with suboptimal doses of 5-FU + iri. alone or in combination with SN-6. First, we confirmed the presence of secreted mucins in the PDO by Alcian blue staining (*Figure 4—figure supplement 1D*). Next, and consistent with our cellular studies, mucin secretion inhibition with SN-6 increased up to 40 times the sensitivity of PDO cells to 5-FU + iri. compared with cells treated with vehicle (IC50 5-FU + iri. + vehicle = 0.90 µg/mL, IC50 5-FU + iri. + SN-6 = 0.02 µg/mL) (*Figure 4D*).

Together, these results show that mucin secretion inhibitors alleviate the resistance to chemotherapeutics agents of cells that secrete mucins.

## Discussion

Mucins are high-molecular-weight O-glycosylated proteins produced by goblet cells that line the epithelium of several organs of the airway and the digestive tract (*Adler et al., 2013*; *Pelaseyed et al., 2014*). These proteins are the major macrocomponents of the mucus layer, which is a physical barrier that protects the underlying tissue from external insults (*Cone, 2009*). Importantly, the combination of different mucins and the type of secretion (baseline or stimulated) modulates the properties of the mucus layer to meet the tissue demands. For instance, baseline secretion is the prominent pathway that maintains the colonic mucus layer, while stimulated is the main pathway to rapidly protect the epithelium after allergen or toxic challenge in the airways (*Zhu et al., 2015*). Accordingly, tight regulation of mucin quantities and composition is essential for the correct functioning of the mucus. Alterations in their synthesis or release are, therefore, hallmarks of several pathological situations like severe asthma or CRC. Hyperproduction of mucin in some subtypes of CRC leads to enhanced chemoresistance of cancer cells, thereby reducing patients' survival (*Pothuraju et al., 2020*; *Jin et al., 2017*). However, in some cases, decreased expression of secreted mucins (i.e. MUC2 and MUC5AC) is an indicator of more aggressive colorectal tumour, possibly because these cells are more dedifferentiated and thus more invasive (*Luo et al., 2019*; *Kocer et al., 2002*). Interestingly, inhibition of O-glycosylation renders pancreatic cancers overexpressing the transmembrane mucin MUC1 more sensitive to 5-FU (*Kalra and Campbell, 2007*). Transmembrane mucins contribute to chemoresistance by inhibiting apoptosis or promoting epithelial to mesenchymal transition (EMT) (*van Putten and Strijbis, 2017*; *Jonckheere et al., 2014*). But the role of secreted mucins in chemoresistance to CRC has not been explored. The chemicals and proteins employed here affect both MUC2 and MUC5AC

as shown previously (*Cantero-Recasens et al., 2019*; *Cantero-Recasens et al., 2018*), but for the sake of simplicity we have monitored only the effects on MUC5AC secretion.

## Cancer cells produce mucins as a protective response to chemotherapy

Our data show that CRC cells produce mucins in response to 5-FU + iri., a first-choice chemotherapy for most CRC (*Kamnerdsupaphon et al., 2007*). Besides, 5-FU + iri. not only increases mucin production, but also increases mucin secretion (70–80% after 6 hr). This increase in secretion is completely blocked by SN-6 treatment (as described previously in *Cantero-Recasens et al., 2019*). This is extremely important because secreted mucins can create a physical barrier that could prevent drugs (for instance, chemotherapeutics) from reaching the tumour cells. Another possibility is that 5-FU + iri. is actively retained by mucin fibres. The availability of tagged versions of these chemicals will help to address this issue in the future.

Our results in PDOs and cell lines show that cancer cells respond to chemotherapeutics by secreting copious amounts of mucin to form a barrier against the treatment. We suggest that this reaction (mucin secretion) of CRC cells to chemotherapy is similar to the programmed response of epithelial cells to toxics or pathogens. In this situation, which could resemble the treatment with 5-FU + iri., mucin-producing cells would release large quantities of mucins to isolate them from the insults (toxins, allergens, or pathogens) (*Adler et al., 2013*). An interesting question is how 5-FU + iri. triggers mucin secretion and whether there is a specific receptor involved in this response or an intracellular pathway that is activated by these chemicals. This finding is important in the application of chemotherapeutics and merits further investigation. Our results show that this increase in mucin production/secretion triggered by chemotherapeutics is independent of transcriptional effects on mucins, although we cannot completely discard the possibility that the levels of components necessary for mucin modification and secretion (e.g., glycosyltransferases) are altered.

## KChIP3, a novel risk factor for mucinous CRC

We postulated that mucin levels could be used as prognostic biomarkers for CRC. Our gene-expression analyses of a public CRC cohort, however, revealed that mucin levels do not reflect the prognosis of CRC patients. Although surprising, this is not totally unexpected because mucin secretion is an extremely well-regulated process to prevent pathological situations. Therefore, high intracellular levels do not directly translate to increased extracellular levels. Our gene-expression data reveal that low level of KChIP3, which negatively regulates baseline secretion (*Cantero-Recasens et al., 2018*), is an important risk factor for untreated tumours with high MUC5AC expression levels. We have found that cells with low levels of KChIP3 (that corresponds to more mucin secretion) are more resistant to 5-FU + iri. To summarize, low levels of KChIP3 promote mucin secretion, which is even more dramatic in tumours with high expression of MUC5AC. Increased mucin secretion protects cancer cells from chemotherapy, which drastically reduce CRC patients' DFS. KChIP3 thus emerges as a good prognostic marker, although it may have low relevance as a therapeutic target because it necessitates the use of gene therapy to enhance its expression and block mucin secretion. An exciting option is to target factors that control KChIP3 activity, such as ryanodine receptors or quenchers of intracellular calcium (*Cantero-Recasens et al., 2018*).

## Combined treatment with NCX inhibitors improves chemotherapy efficacy

Mucinous cancers present higher chemoresistance than other subtypes of CRC (*Luo et al., 2019*). As suggested by several authors and now shown by our experimental data, secreted mucins can act as a barrier to external insults including chemotherapeutic drugs. In fact, genetic regulation of this secretion by modulating KChIP3 levels renders cancer cells more sensitive to treatment. However, our results also demonstrate that 5-FU + iri. is a stimulus for mucin secretion, which may be an adaptive response of cancer cells for isolation from harmful environment. This unexpected effect can explain how some cancer cells become more resistant to treatment over time.

Stimulated mucin secretion, by colonic or airways cells, depends on extracellular calcium entry through the reverse mode of NCX (*Mitrovic et al., 2013*; *Cantero-Recasens et al., 2019*). Our data show that inhibition of stimulated mucin secretion reduces the resistance of tumour cells to anticancer treatments. Both NCX inhibitors (benzamil and SN-6) prevent the effect of 5-FU + iri. on mucin

secretion. On the other hand, co-treatment with NCX inhibitors improves the apoptotic effect of 5-FU + iri. on cancer cells, even at low 5-FU + iri. concentrations (IC20). However, one of the main problems of NCX inhibitors is their low specificity and the effects on cardiac function. To avoid these problems, SN-6 was developed as a specific inhibitor of NCX reverse mode that at working concentration does not affect other receptors, transporters, or ion channels, and has less side effects than benzamil (*Niu et al., 2007*; *Iwamoto et al., 2004*). Our results reveal the potential therapeutic importance of SN-6. Finally, it is satisfying to note that the effect of SN-6 on the chemotherapy sensitivity of cancer cells is replicated in PDOs.

## Conclusions

In conclusion, CRC cells produce mucins to create a protective environment against chemotherapy treatments. Blocking mucin secretion by genetic (altering KChIP3 levels) or chemical means (using NCX inhibitors) renders tumour cells more sensitive to anticancer therapies. In addition, we have identified the mucin secretion regulator KChIP3 as a novel risk factor for mucinous CRC, which could be used as a prognostic biomarker. We have provided potential new pharmacological strategies to control chemorefraction of mucinous CRCs, but this approach is likely to benefit patients with mucinous cancers in general.

# Materials and methods

**Key resources table**

| Reagent type (species) or resource | Designation | Source or reference | Identifiers | Additional information |
|---|---|---|---|---|
| Gene (*Homo sapiens*) | MUC5AC | Ensembl | ENSG00000215182 | |
| Gene (*H. sapiens*) | MUC2 | Ensembl | ENSG00000198788 | |
| Gene (*H. sapiens*) | MUC6 | Ensembl | ENSG00000184956 | |
| Gene (*H. sapiens*) | MUC5B | Ensembl | ENSG00000117983 | |
| Gene (*H. sapiens*) | MUC19 | Ensembl | ENSG00000205592 | |
| Gene (*H. sapiens*) | KCNIP3 | Ensembl | ENSG00000115041 | |
| Cell line (*H. sapiens*) | HT29-M6 | ATCC | CVCL_G077 | Mycoplasma free |
| Cell line (*H. sapiens*) | HT29-18N2 | ATCC | CVCL_5942 | Mycoplasma free |
| Antibody | Anti-MUC5AC (mouse monoclonal) | Neomarkers, Waltham, MA | Clone 45M1 | (1:1000) |
| Antibody | Anti-γH2A.X (mouse monoclonal) | Cell Signaling | #2577 | (1:1000) |
| Commercial assay or kit | CometAssay Trevigen Kit | Trevigen | 250-050K | |
| Chemical compound, drug | SN-6 | Sigma-Aldrich | SML1937-5MG | (5 µM) |
| Chemical compound, drug | Benzamil | Sigma-Aldrich | B2417-10MG | 5 µM |

## Bioinformatics analysis

Transcriptomic and available clinical data dataset from CRC was downloaded from the open-access resource CANCERTOOL. We used the Jorissen cohort (GSE14333), which includes expression and clinical data for 226 patients (*Jorissen et al., 2009*). Patients were classified according to the mean expression of either MUC5AC, MUC2, or KChIP3. The association with relapse was assessed using Kaplan–Meier estimates and Cox proportional hazard models. A standard log-rank test was applied to assess significance between groups. This test was selected because it assumes the randomness of the possible censorship. All the survival analyses and graphs were performed with R using the survival (v.3.2-3) and survimer (v.0.4.8) packages, and a p-value<0.05 was considered statistically significant.

## Reagents and antibodies

All chemicals were obtained from Sigma-Aldrich (St. Louis, MO) except anti-MUC5AC antibody clone 45M1 (Neomarkers, Waltham, MA). Secondary antibodies for immunofluorescence microscopy were from Life Technologies.

## Cell lines

### HT29-M6

HT29-M6 cells (obtained from ATCC; RRID: CVCL_G077) were grown in Dulbecco's modified Eagle's medium (Invitrogen) plus 10% foetal bovine serum (Biological Industries) and were maintained in 5% $CO_2$ incubator at 37°C. All experiments were performed with cells at 6 days post-confluency when they form a well-polarized monolayer and present higher levels of mucins.

### HT29-18N2

HT29-18N2 cells (obtained from ATCC) (RRID:CVCL_5942) were tested for mycoplasma contamination with the Lookout mycoplasma PCR detection kit (Sigma-Aldrich). Mycoplasma-negative HT29-18N2 cells were used for the experiments presented here. HT29-18N2 cells were differentiated to goblet cells as described previously (*Mitrovic et al., 2013*) Briefly, cells were seeded in complete growth medium (DMEM complemented with 10% FCS and 1% P/S), and the following day (day zero: D-0), the cells were placed in PFHMII protein-free medium (Gibco, Thermo Fisher Scientific, Waltham, MA). After 3 days (D-3), medium was replaced with fresh PFHMII medium and cells grown for three additional days. At day 6 (D-6), cells were trypsinized and seeded for the respective experiments in complete growth medium followed by incubation in PFHMII medium at day 7 (D-7). All experimental procedures were performed at day 9 (D-9).

## Patient-derived organoid

Samples from patients were kindly provided by MARBiobank and IdiPAZ Biobank, integrated in the Spanish Hospital Biobanks Network (RetBioH; http://www.redbiobancos.es/). Informed consent was obtained from all participants, and protocols were approved by institutional ethical committees. The project was approved by the ethical committee of the Hospital del Mar (Barcelona, Spain) (REF 2019/8595/I). For the PDO generation, primary tumour from was disaggregated in 1.5 mg/mL collagenase II (Sigma) and 20 µg/mL hyaluronidase (Sigma) after 40 min of incubation at 37°C, filtered in 100 µm cell strainer and seeded in 50 µL Matrigel in 24-well plates, as previously described (*Sato et al., 2011*). After polymerization, 450 µL of complete medium was added (DMEM/F12 plus penicillin [100 U/mL] and streptomycin [100 µg/mL], 100 µg/mL Primocin, 1X N2 and B27, 10 mM nicotinamide; 1.25 mM N-acetyl-L-cysteine, 100 ng/mL Noggin, and 100 ng/mL R-spondin-1, 10 µM Y-27632, 10 nM PGE2, 3 µM SB202190, 0.5 µM A-8301, 50 ng/mL EGF and 10 nM Gastrin I). Tumour spheres were collected and digested with an adequate amount of trypsin to single cells and re-plated in culture. Cultures were maintained at 37°C, 5% $CO_2$ and medium changed every week. For dose–response curves, 600 single PDO cells were plated in 96-well plates in 10 µL Matrigel with 100 µL of complete medium. After 6 days in culture, the growing PDO was treated with combinations of 5-FU + iri. plus SN-6 or 5-FU + iri. After 72 hr of treatment, we measured the cell viability using the CellTiter-Glo 3D Cell Viability Assay following the manufacturer's instructions in an Orion II multiplate luminometer. The mutation identified in the PDO used in this study was KRAS G12D (66.43%).

## Survival curves

HT29-18N2 cells (KD/OV) were seeded in 96-well plates with Dulbecco's modified Eagle's medium. Confluent cells were treated with combination of 5-FU + iri. for 72 hr at indicated concentrations. Cell viability was determined using the CellTiter-Glo 3D Cell Viability Assay (Promega) following the manufacturer's instructions in an Orion II multiplate luminometer (Berthold detection systems).

## qRT-PCR analysis

Total RNA was extracted with the RNeasy Mini Kit (QIAGEN ref 74004), and the RT-First Strand cDNA Synthesis Kit (GE Healthcare Life Sciences ref 27-9261-01) was used to produce cDNA. qRT-PCR was performed in LightCycler 480 system using SYBR Green I Master Kit (Roche ref 04887352001). Primers

**Table 1.** Primer sequences used for detecting mRNA for the respective genes.

| Gene | Forward primer (5′–3′) | Reverse primer (5′–3′) |
| --- | --- | --- |
| MUC5AC | CTGGTGCTGAAGAGGGTCAT | CAACCCCTCCTACTGCTACG |
| TBP | TGCCCGAAACGCCGAATATAATC | GTCTGGACTGTTCTTCACTCTTGG |
| GAPDH | GTCATCCCTGAGCTGAACG | CTCCTTGGAGGCCATGTG |

for each gene (sequences shown in *Table 1*) were designed using Primer-BLAST (NCBI) (*Ye et al., 2012*), limiting the target size to 150 bp and the annealing temperature to 60°C.

## Secretion assay and Western blot

HT29 cells were grown to confluency and, 7 days post-confluency, cells were incubated with 5-FU + iri. (25 µg/mL 5-Fu + 10 µg/mL irinotecan in normal tissue culture medium) alone or in combination with mucin secretion inhibitor SN-6. At the same time, extracellular medium was collected for all time points and centrifuged for 5 min at 800 × *g* at 4°C. Cells were washed and then lysed in 500 µL of PBS plus 0.5% Triton X-100, 1 mM EDTA, 100 mM Na-orthovanadate, 0.25 mM phenylmethylsulfonyl fluoride, and cOmplete Protease Inhibitor Cocktail (Roche) for 20 min at 4°C (scheme in *Figure 1—figure supplement 1A* and *Figure 4—figure supplement 1A*). Extracellular medium and lysates were analysed by WB using standard SDS-polyacrylamide gel electrophoresis (SDS-PAGE) techniques.

Briefly, lysates were boiled in Laemmli buffer, run in polyacrylamide gels (8% for tubulin, 15% for histones), and transferred to polyvinylidene fluoride (PVDF) membranes (Millipore). To check mucin levels in the secreted medium, reducing buffer was added to collected medium and samples were boiled for 10 min. When testing MUC5AC levels, protein labels were prepared in a reducing buffer and run in 4% acrylamide gels. The membranes were incubated overnight at 4°C with the appropriate primary antibodies. After washing, the membranes were incubated with specific secondary horseradish peroxidase-conjugated antibodies (Dako, Denmark) and visualized using the enhanced chemiluminescence reagent from Amersham. Gel bands were quantified using the Measurement Log function from Adobe Photoshop CC.

Uncropped gels presented in *Figures 1 and 4* are provided as source data (*Figure 1—source data 1* and *Figure 4—source data 1*).

## Comet assay

HT29-M6 cells were plated in 12-well plates with Dulbecco's modified Eagle's medium. 4 days after arriving to 100% confluency, cells were treated with 5-FU + iri. (25 µg/mL 5-Fu + 10 µg/mL irinotecan) alone or in combination with secretion inhibitor benzamil 20 µM for 72 hr. Comet assays were performed using CometAssay Trevigen Kit (4250-050K) following the manufacturer's instructions. Pictures were taken using a Nikon Eclipse Ni-E epifluorescence microscope, and tail moment was calculated using the OPENCOMET plugin for ImageJ.

## Cell viability assay

HT29-M6 cells were plated in 6-well plates with Dulbecco's modified Eagle's medium. After 4 days in post-confluency, CRC cells were treated with 5-FU + iri. (10 µg/mL 5-Fu + 4 µg/mL irinotecan) alone or in combination with secretion inhibitor benzamil 20 µM. Cell viability was determined by flow cytometry 24 hr or 72 hr after treatment, staining cells with DAPI (5 µg/mL) and measuring alive cells in the LSR-Fortessa analyzer.

## Microscopy and colocalization analysis

HT29-M6 cells were plated in 6-well plates with Dulbecco's modified Eagle's medium. After being 7 days at post-confluency, cells were treated with 5-FU + iri. (50 µg/mL 5-Fu + 20 µg/mL irinotecan) alone or in combination with secretor inhibitor SN-6 10 µM for 24 hr. Then cells were directly fixed with 4% paraformaldehyde, permeabilized with 0.3% Triton X-100 (Pierce), washed and incubated overnight with the corresponding primary antibodies. The samples were developed with TSA Plus Cyanine 3/Fluorescein System (PerkinElmer) and mounted in ProLong Diamond Antifade Mountant plus DAPI (Thermo Scientific).

Images were acquired using an inverted Leica SP8 or Leica SP5 confocal microscope with a ×63 Plan Apo NA 1.4 objective and analysed using ImageJ (Fiji, version 2.0.0-rc-43/1.51g) (*Schindelin et al., 2012*). For detection of the respective fluorescence emission, the following laser lines were applied: DAPI, 405 nm; and Alexa Fluor 555, 561 nm; Alexa Fluor 647, 647 nm. Two-channel colocalization analysis was performed using ImageJ, and the Manders' correlation coefficient was calculated using the plugin JaCop (*Bolte and Cordelières, 2006*).

## Alcian blue staining

Previously PDOs de-paraffinized sections or 4% PFA-fixed HT29-M6 cells were incubated with 3% acetic acid in $H_2O$ for 3 min, immersed in Alcian blue (10 mg/mL pH 2.5) (Merck ref 101647) for 1 hr, and counterstained with Nuclear Fast Red solution (Sigma ref 6409-77-4) for 10 min. Samples were mounted in DPX mountant (Sigma ref 06522), and images were obtained with an Olympus BX61 microscope.

## Statistical analysis

All data are means ± SEM of a minimum of three biological replicates (N ≥ 3). In all cases, a D'Agostino–Pearson omnibus normality test was performed before any hypothesis contrast test. Statistical analysis and graphics were performed using GraphPad Prism 6 (RRID:SCR_002798). For data that followed normal distributions, we applied either Student's *t*-test or one-way analysis of variance (ANOVA) followed by Tukey's post hoc test. For data that did not fit a normal distribution, we used Mann–Whitney's unpaired *t*-test and nonparametric ANOVA (Kruskal–Wallis) followed by Dunn's post hoc test. Criteria for a significant statistical difference were *$p < 0.05$; **$p < 0.01$.

## Acknowledgements

We thank all members of the Malhotra and Espinosa Lab for valuable discussions. Fluorescence microscopy was performed at the Advanced Light Microscopy Unit at the CRG, Barcelona. VM is an Institució Catalana de Recerca i Estudis Avançats professor at the Centre for Genomic Regulation. This work was funded by grants from the Spanish Ministry of Economy and Competitiveness (BFU2013-44188-P to VM) and FEDER Funds, and Instituto de Salud Carlos III (PI19-00013 to LE). We acknowledge support of the Spanish Ministry of Economy and Competitiveness, through the Programmes 'Centro de Excelencia Severo Ochoa 2013–2017' (SEV-2012-0208 and SEV-2013-0347) and Maria de Maeztu Units of Excellence in R&D (MDM-2015-0502). This work reflects only the authors' views, and the EU Community is not liable for any use that may be made of the information contained therein.

## Additional information

### Competing interests

Vivek Malhotra: Senior editor, *eLife*. The other authors declare that no competing interests exist.

### Funding

| Funder | Grant reference number | Author |
| --- | --- | --- |
| Ministerio de Economía, Industria y Competitividad, Gobierno de España | BFU2013-44188-P | Vivek Malhotra |
| Instituto de Salud Carlos III | PI19-00013 | Lluis Espinosa |

The funders had no role in study design, data collection and interpretation, or the decision to submit the work for publication.

### Author contributions

Gerard Cantero-Recasens, Conceptualization, Formal analysis, Investigation, Methodology, Validation, Writing – original draft, Writing – review and editing; Josune Alonso-Marañón, Investigation, Methodology, Validation, Writing – review and editing; Teresa Lobo-Jarne, Formal analysis, Investigation,

Methodology, Validation, Visualization, Writing – original draft, Writing – review and editing; Marta Garrido, Investigation, Methodology, Writing – review and editing; Mar Iglesias, Conceptualization, Resources, Writing – review and editing; Lluis Espinosa, Conceptualization, Formal analysis, Funding acquisition, Investigation, Resources, Supervision, Writing – original draft, Writing – review and editing; Vivek Malhotra, Conceptualization, Formal analysis, Funding acquisition, Resources, Supervision, Writing – original draft, Writing – review and editing

### Author ORCIDs
Gerard Cantero-Recasens http://orcid.org/0000-0001-6452-782X
Lluis Espinosa http://orcid.org/0000-0002-2897-4099
Vivek Malhotra http://orcid.org/0000-0001-6198-7943

### Decision letter and Author response
Decision letter https://doi.org/10.7554/eLife.73926.sa1
Author response https://doi.org/10.7554/eLife.73926.sa2

## Additional files

### Supplementary files
• Transparent reporting form

### Data availability
All data generated or analysed are included in the manuscript.

The following dataset was generated:

| Author(s) | Year | Dataset title | Dataset URL | Database and Identifier |
|---|---|---|---|---|
| Jorissen RN, Gibbs P, Christie M, Prakash S, Lipton L, Desai J, Kerr D, Aaltonen LA, Arango D, Kruhøffer M, Orntoft TF, Andersen L, Gruidl M, Kamath VP, Eschrich S, Yeatman TJ, Sieber OM | 2009 | Jorissen Cohort | https://www.ncbi.nlm.nih.gov/geo/query/acc.cgi?acc=GSE14333 | NCBI Gene Expression Omnibus, GSE14333 |

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
