## [Editor Report]

Hypersecretion of mucins by colorectal cancer (CRC) cells confers resistance to immune surveillance, and this phenomenon has also been postulated to confer resistance to chemotherapy. This work describes studies aimed at investigating mucus secretion and proteins influencing mucus secretion and the impact on the efficacy of a commonly used chemotherapy treatment. These results shed light on the chemoresistance of mucinous tumors and point to a possible prognostic marker involved in mucus secretion.

---

## [Decision Letter]

**Decision letter after peer review:**

Thank you for submitting your article "Reversing chemorefraction in colorectal cancer cells by controlling mucin secretion" for consideration by *eLife*. Your article has been reviewed by 2 peer reviewers, and the evaluation has been overseen by a Reviewing Editor and Suzanne Pfeffer as the Senior Editor. The following individual involved in review of your submission has agreed to reveal their identity: Benjamin S Glick (Reviewer #2).

The reviewers have discussed their reviews with one another and essential points are listed below.

Essential revisions:

1. Additional data supporting mucus secretion as a mechanism for effects of chemotherapy would be necessary to support the main conclusions of the paper. To conclude an inhibitory effect of secreted mucus on the accessibility of the drugs, mucus secretion and reduced drug accessibility by mucus must be demonstrated by relevant analyses.

2. Regarding chemotherapeutic drugs increasing mucin secretion: it is obvious why that response aids the tumor cells, but not obvious why tumor cells are programmed to respond in that way. Please address this question.

*Reviewer #1 (Recommendations for the authors):*

Some suggestions mostly linked to the points raised in the public review that could improve the study.

The contribution from other mucins in the CRC survival analysis and in the cell expression and secretion studies would be of importance as the impact of them might influence the outcome. Complementing any staining of intracellular and secreted mucus with a general mucin staining approach (possibly by utilizing the glycans) would possibly give a more comprehensive view.

The mucus secretion in cell lines will be impacted by cell proliferation, survival and polarization that are all important factors to monitor and show or discuss to reveal how they were accounted for.

To conclude an inhibitory effect of secreted mucus on the accessibility of the drugs, mucus secretion and reduced drug accessibility by mucus must be demonstrated by relevant analyses. Mucus secretion is preferably assessed by complementary methods monitoring intracellular vesicles using microscopy and measuring secreted mucus using dotblot, a method previously performed by the group. As the FOLFIRI treatment induce cell death, contamination of mucus from dead or detached cells will contribute, but a relative comparison to the amount of any general intracellular protein in the samples could shed light on the size of this fraction. Another option is to separate the secreted reduced mucus on more porous gel-types to estimate the amount of the not fully glycosylated material compared to the fully glycosylated secreted mucins. The effect on transcript expression would add a nice complement to these studies deciphering the transcriptional element of mucus production. Assessment of mucus secretion upon modulation of the FOLFIRI effect by KChipP3 expression levels or Benzamil and SN-6 treatment would be required to provide a functional link to mucus.

Assessing mucus release in the organoids would strengthen the argument substantially. Mucus staining of fixed samples could be explored.

As the overall basis of the study is the link between mucin secretion and CRC treatment efficacy it would be good if the patient material could be stratified by treatment with separation of groups based on mucin expression and the related genes.

The rationale for using two different HT-29 clones should be explained.

*Reviewer #2 (Recommendations for the authors):*

1. In Figure 1 and other figures, the abbreviation "FOLFIRI" is apparently used in place of "5-FU+iri". The authors should choose one or the other and be consistent.

2. In Figure 1C, it's hard to tell what we are looking at with albumin. Are endosomes being visualized in the washed cells? What are the labeled structures in the cells with mucin fibers? In Figure 1D, why would albumin overlap with DAPI?

3. The abbreviation "HR" should be defined. "DSF" in the text should probably be "DFS" as defined in the subsequent sentence. And if "CT" stands for chemotherapy, this abbreviation should be either defined or avoided.

4.In Figure 2C, is there truly no significant difference? By 100 months, the two populations look radically different. Either something is wrong with the statistics, or the analysis has a problem.

5. The description of Figure 3B is confusing. The authors seem to say that the combination treatment is effective, then not effective, then effective again. Is the point that the effect becomes visible only after 72 h?

6. Figure 4A and 4C seem to be contradictory, because Figure 4A shows no significant increase in (g)H2A.X with SN-6 whereas Figure 4C shows a strong increase. Please clarify.

[Editors' note: further revisions were suggested prior to acceptance, as described below.]

Thank you for resubmitting your work entitled "Reversing chemorefraction in colorectal cancer cells by controlling mucin secretion" for further consideration by *eLife*. Your revised article has been evaluated by Suzanne Pfeffer (Senior Editor) and a Reviewing Editor.

The manuscript has been improved but there are some remaining issues that need to be addressed, as outlined below:

In light of the comments by Reviewer #1 below, the following revisions are requested:

1. Given that the model is that 5FU+iri increases mucin secretion, direct demonstration of increased mucin secretion (as opposed to mucin vesicle production) upon 5FU+iri treatment vs. vehicle control should be shown. An experiment similar to what is shown in Figure 1C but with 5FU+iri treatment vs vehicle would address this. Also, westerns of Muc5AC from media and cell lysates of vehicle treated vs. 5FU+iri treated (similar to what is shown in Figure 4A) would allow quantitation of this increase and would strengthen the paper.

2. There is still some confusion over the link between Kchip3 expression, mucin secretion, chemo treatment and CRC patient outcome. The authors presumably already have images showing increased and decreased mucin secretion in the cell lines where kchip3 has been mutated or overexpressed relative to controls-showing these data would greatly strengthen the paper. Additionally, if the authors could include some clarification/discussion that the patient data is based on gene expression rather than mucin secretion, and does not include information on chemotherapy treatment, this may resolve some of the confusion.

*Reviewer #1 (Recommendations for the authors):*

The authors have addressed several of the concerns raised and show nicely that the secreted material does not contain dead or detached cells. They also provide important information on mucin expression upon treatment of cells although the effect of the inhibitor SN6 alone in reducing mucin expression is not discussed. The main issue with secretion is addressed in relation to 5-FU+Iri and SN6 treatment, but not to the most central component of the study, KChiP3, relevant data is however referred to. The secreted mucins are detected by WB at a size of above 450 kDa and as the theoretical size of the naked protein in the range of 580 kDa it indicates to be the secreted product, however with limited glycosylation (a gel stained by Alcian blue would show this). However, the discrepancy in the data on mucus secretion partly remains as the results in Figure 1B, 4A and 4B show enhanced intracellular MUC5AC upon 5-FU+Iri treatment (referred to as production as it is not dependent on expression), but the data in Figure 3C with reduced number and size of vesicles is not in line. Mucus secretion was further analyzed using general staining (Alcian blue) of treated cells and organoids. The data on cells is very hard to interpret while the stained organoid shows secreted mucus. In this context treatment of the organoid with 5-FU+Iri would have been a much better representation of enhanced secretion upon treatment compared to the fixed cells.

The link between enhanced mucin expression leading poor prognosis for CRC patients and reduced effects of treatment depending on mucus secretion is explained by differentiation, but this is a main concern in the concept and could benefit from even further discussion.*Reviewer #2 (Recommendations for the authors):*

The authors have done a satisfactory job of responding to my previous comments. I support publication of the revised manuscript.

[Editors' note: further revisions were suggested prior to acceptance, as described below.]

Thank you for resubmitting your work entitled "Reversing chemorefraction in colorectal cancer cells by controlling mucin secretion" for further consideration by *eLife*. Your revised article has been evaluated by Suzanne Pfeffer (Senior Editor) and a Reviewing Editor.

We are pleased to notify you that it is now acceptable for publication, provided that the authors include numbers of samples and replicates for all figures (as Figure 4A) and a brief explanation, as given in the response letter, why there is a variation observed in Figure 3C compared to other results presented.

*Reviewer #1 (Recommendations for the authors):*

The manuscript is now improved by text alterations. It is unfortunate that no extra data is provided proving mucus secretion to affect CRC treatment, but the text now includes most of the information needed for proper evaluation by readers. In this context I would find it important to include numbers of samples and replicates for all figures (as Figure 4A) and to include a brief explanation, as given in the response letter, why there is a variation observed in Figure 3C compared to other results presented to be fully transparent. With these additions I do find it acceptable for publication.

---

## [Author Response]

Essential revisions:1. Additional data supporting mucus secretion as a mechanism for effects of chemotherapy would be necessary to support the main conclusions of the paper. To conclude an inhibitory effect of secreted mucus on the accessibility of the drugs, mucus secretion and reduced drug accessibility by mucus must be demonstrated by relevant analyses.

We have included new data to support that secretion of mucins is an important mechanism for enhanced resistance to chemotherapy. Our results show that mucins act as a physical barrier to prevent drug accessibility. In the revised version, we provide new data showing extracellular mucus (staining with Alcian Blue) in cells and PDOs (Figure 4 – Supplement figure 1A and 1C) and show that this effect is independent of transcriptional regulation of mucins (Figure 4 – Supplement figure 1B). In addition, we have added data on mucin secretion by treatment with chemotherapy and NCX inhibitor SN-6 (Figure 4A). These data are now included in the Results section. We also discuss the contribution of secreted mucins on the accessibility of drugs as stated below.

“Cancer cells produce mucins as a protective response to chemotherapy

Our data show that CRC cells produce mucins in response to 5-Fluorouracil + Irinotecan (5-FU+iri.), a first-choice chemotherapy for most CRC (30). Besides, 5-FU+iri. not only increases mucin production, but also increases mucin secretion (70-80% after 6 hours). This increase in secretion is completely blocked by SN-6 treatment (as described previously in (6)). This is extremely important, because secreted mucins can create a physical barrier that could prevent drugs (for instance, chemotherapeutics) from reaching the tumour cells. Another possibility is that 5-FU+iri. is actively retained by mucin fibres. The availability of tagged versions of these chemicals will help to address this issue in the future.

Our results in PDOs and cell lines show that cancer cells respond to chemotherapeutics by secreting copious amounts of mucin to form a barrier against the treatment. We suggest that this reaction (mucin secretion) of colorectal cancer cells to chemotherapy is similar to the programmed response of epithelial cells to toxics or pathogens. In this situation, which could resemble the treatment with 5-FU+iri., mucin-producing cells would release large quantities of mucins to isolate them from the insults (toxins, allergens or pathogens) (3). An interesting question is how 5-FU+iri. triggers mucin secretion and whether there is a specific receptor involved in this response or an intracellular pathway that is activated by these chemicals. This finding is important in application of chemotherapeutics and merits further investigation. Our results show that this increase in mucin production/secretion triggered by chemotherapeutics is independent of transcriptional effects on mucins, although we cannot completely discard the possibility that the levels of components necessary for mucin modification and secretion (e.g., glycosyltransferases) are altered.”

2. Regarding chemotherapeutic drugs increasing mucin secretion: it is obvious why that response aids the tumor cells, but not obvious why tumor cells are programmed to respond in that way. Please address this question.

This is an interesting question. We suggest that this is a programmed response of epithelial cells to toxics or pathogens. In this situation (similar to the change in the environment that cancer cells detect after chemotherapy treatment) mucin-producing cells release huge amounts of mucins to isolate them from the toxins, pathogens or allergens. Interestingly, our new results show that this increase in mucin production and secretion is independent of transcriptional effects (Figure 4 – Supplement figure 1B). We have extended the previous paragraph on this topic in the discussion:

“We suggest that this reaction (mucin secretion) of colorectal cancer cells to chemotherapy is similar to the programmed response of epithelial cells to toxics or pathogens. In this situation, which could resemble the treatment with 5-FU+iri., mucin-producing cells would release large quantities of mucins to isolate them from the insults (toxins, allergens or pathogens) (3). An interesting question is how 5-FU+iri. triggers mucin secretion and whether there is a specific receptor involved in this response or an intracellular pathway that is activated by these chemicals. This finding is important in application of chemotherapeutics and merits further investigation. Our results show that this increase in mucin production/secretion triggered by chemotherapeutics is independent of transcriptional effects on mucins, although we cannot completely discard the possibility that the levels of components necessary for mucin modification and secretion (e.g., glycosyltransferases) are altered.”

We have also included the following for the analysis of MUC5AC transcription in the Results section:

“In addition, to test for the transcriptional effect of SN-6 on 5-FU+iri. dependent mucin production, we analysed RNA levels of MUC5AC in HT29-M6 cells treated with vehicle (control), 5-FU+iri. (IC50), 10 µM SN-6 or 5-FU+iri. + SN-6. Our results revealed that the increase in mucin production and secretion by 5-FU+iri. and the effect of SN-6 on 5-FU+iri. is independent of MUC5AC transcription (Figure 4 – Supplement figure 1B).”

Reviewer #1 (Recommendations for the authors):Some suggestions mostly linked to the points raised in the public review that could improve the study.The contribution from other mucins in the CRC survival analysis and in the cell expression and secretion studies would be of importance as the impact of them might influence the outcome. Complementing any staining of intracellular and secreted mucus with a general mucin staining approach (possibly by utilizing the glycans) would possibly give a more comprehensive view.

As described above, now we include staining of the PDOs and HT29 cells with Alcian-Blue to show the mucus (Figure 4 – Supplement figure 1A and 1C).

The mucus secretion in cell lines will be impacted by cell proliferation, survival and polarization that are all important factors to monitor and show or discuss to reveal how they were accounted for.

All experiments were performed at six days of post confluency, when HT29 form a polarized monolayer. As shown in the immunofluorescence images, there are no major differences in cell proliferation and survival between conditions. Nevertheless, same amounts of protein were loaded in the WB for each condition, and cell markers (i.e., tubulin) included in each experiment to normalize the data.

To conclude an inhibitory effect of secreted mucus on the accessibility of the drugs, mucus secretion and reduced drug accessibility by mucus must be demonstrated by relevant analyses. Mucus secretion is preferably assessed by complementary methods monitoring intracellular vesicles using microscopy and measuring secreted mucus using dotblot, a method previously performed by the group. As the FOLFIRI treatment induce cell death, contamination of mucus from dead or detached cells will contribute, but a relative comparison to the amount of any general intracellular protein in the samples could shed light on the size of this fraction. Another option is to separate the secreted reduced mucus on more porous gel-types to estimate the amount of the not fully glycosylated material compared to the fully glycosylated secreted mucins.

As the reviewer suggested before, we now include data showing that the secreted fraction does not contain cytosolic proteins to exclude contribution of cell lysis (Figure 4A).

The effect on transcript expression would add a nice complement to these studies deciphering the transcriptional element of mucus production.

We have included new data (qPCR) on the effect on transcriptional regulation of mucins (Figure 4 – Supplement figure 1B).

Assessment of mucus secretion upon modulation of the FOLFIRI effect by KChipP3 expression levels or Benzamil and SN-6 treatment would be required to provide a functional link to mucus.

As the reviewer suggested, we have assessed the mucin secretion after FOLFIRI treatment (Figure 4A). Our results show that FOLFIRI treatment not only increase production, but also secretion of mucins. Importantly, treatment with NCX inhibitor SN-6 blocks this secretion, which provides a link between the treatment and mucins.

Assessing mucus release in the organoids would strengthen the argument substantially. Mucus staining of fixed samples could be explored.

We have stained PDOs and cells with Alcian Blue to detect extracellular mucus (Figure 4 – Supplement figure 1A and 1C).

As the overall basis of the study is the link between mucin secretion and CRC treatment efficacy it would be good if the patient material could be stratified by treatment with separation of groups based on mucin expression and the related genes.

Current chemotherapy treatments are very standardized (mainly 5-FU plus platinum derivatives or irinotecan) with multiple variations and adjusted based on patient response and characteristics. Thus, stratification of patients by treatment (when available) represents the loss of significance.

The rationale for using two different HT-29 clones should be explained.

As stated above, both HT29 clones are derived from the more heterogeneous HT29 cell line and are highly comparable and grow as well-polarized monolayers that differentiate after reaching confluence (Phillips et al., 1995). Therefore, all experiments were done on the same days of post-confluence.

Reviewer #2 (Recommendations for the authors):1. In Figure 1 and other figures, the abbreviation "FOLFIRI" is apparently used in place of "5-FU+iri". The authors should choose one or the other and be consistent.

We have changed all the labels in the figures to 5-FU+iri to be consistent.

2. In Figure 1C, it's hard to tell what we are looking at with albumin. Are endosomes being visualized in the washed cells? What are the labeled structures in the cells with mucin fibers? In Figure 1D, why would albumin overlap with DAPI?

The main point of Figure 1C is to show that mucins function as a physical barrier to external particles. In absence of mucin barrier, albumin will be uptaken by Cubilin-Amnionless (AMN) receptor complex and delivered to endosomes. In cells with mucin fibers, we expect that these structures represent albumin that is not internalized.

We intended to use DAPI-Albumin overlap as a measure of the intracellular uptake, but to avoid any misunderstanding we have removed it.

3. The abbreviation "HR" should be defined. "DSF" in the text should probably be "DFS" as defined in the subsequent sentence. And if "CT" stands for chemotherapy, this abbreviation should be either defined or avoided.

Thanks, we have included the definition of HR, substituted DSF for DFS and removed CT from the text.

4.In Figure 2C, is there truly no significant difference? By 100 months, the two populations look radically different. Either something is wrong with the statistics, or the analysis has a problem.

The statistical analysis in Figure 2C takes into account the whole curve, and since for the first 100 months both curves are very similar, there are no significant differences. Interestingly, we have found three patients that behave differently from the rest and were responsible for this change after 100 months. We have found that these patients have low levels of MUC5AC but high levels of other secreted mucins, which could explain the fall in DFS (Figure 2 – Supplement figure 1A). We now describe this in the discussion as stated below.

“There is no significant effect on patients with low MUC5AC expression (Figure 2C), although after 100 months there is a decrease in DFS. Further analysis of this effect revealed that three patients (GSM358532, GSM358534 and GSM358438) were responsible of this effect. We studied these patients in more detail and interestingly, although they have low levels of MUC5AC, these patients present increased levels of other secreted mucins (e.g., MUC2) (Figure 2 – Supplement figure 1A). However, similarly to previous studies (10, 12), analysis of the effect of MUC2 levels on DFS shows that high MUC2 levels are protective while patients with low levels of MUC2, which may reflect a more dedifferentiated state of cancer cells, have a lower DFS (HR=2.54, p=0.023) (Figure 2 – Supplement figure 1B). Importantly, when we studied the effect of KChIP3 on the DFS of patients with high expression of MUC2, we found a clear tendency (p=0.08) to worse prognosis in patients with low levels of KChIP3 (Figure 2 – Supplement figure 1C). Altogether, these results suggest that low levels of KChIP3, which consequently increase the quantity of mucins secreted, protect cancer cells from chemotherapeutic drugs.”

5. The description of Figure 3B is confusing. The authors seem to say that the combination treatment is effective, then not effective, then effective again. Is the point that the effect becomes visible only after 72 h?

Yes, the point is that the effect becomes visible after 72h. We apologize for this confusion; we have rewritten the sentences to make this consistent as stated below.

“We then tested cell viability of HT29-M6 at 24- and 72-hours post-treatment of 5-FU+iri. alone or in combination with Benzamil. There was no significant effect of the combined Benzamil and 5-FU+iri. treatment at 24 hours. At 72 hours, however, the therapeutic effect of low concentration of 5-FU+iri. (15% reduction on cell viability) was strongly enhanced when combined with 20 µM Benzamil (40% reduction on cell viability compared with the control) (Figure 3B). These results demonstrate that treatment with the NCX inhibitor Benzamil improves the effect of chemotherapy on tumour cell eradication.”

6. Figure 4A and 4C seem to be contradictory, because Figure 4A shows no significant increase in (g)H2A.X with SN-6 whereas Figure 4C shows a strong increase. Please clarify.

In Figure 4A there is some increase in the SN-6 treatment at 1 and 3 hours, which is not so evident at 6 hours likely due to saturation of the signal. Figure 4C is conceptually different as it represents the percent of cells that are positive for (g)H2A.X and not the amount of signal. Despite that, we now include a new Figure 4A with lower amounts of cell lysate, which confirms the increase in yH2A.X.

[Editors' note: further revisions were suggested prior to acceptance, as described below.]

The manuscript has been improved but there are some remaining issues that need to be addressed, as outlined below:In light of the comments by Reviewer #1 below, the following revisions are requested:1. Given that the model is that 5FU+iri increases mucin secretion, direct demonstration of increased mucin secretion (as opposed to mucin vesicle production) upon 5FU+iri treatment vs. vehicle control should be shown. An experiment similar to what is shown in Figure 1C but with 5FU+iri treatment vs vehicle would address this. Also, westerns of Muc5AC from media and cell lysates of vehicle treated vs. 5FU+iri treated (similar to what is shown in Figure 4A) would allow quantitation of this increase and would strengthen the paper.

Data demonstrating that 5-FU+iri promotes mucin secretion is already shown in Figure 4A. It is important to state that 5-FU+iri is diluted in tissue culture medium, so the comparison of time 0 vs. the other time points is the experiment that the reviewer is requesting. In more detail, cells are grown to confluency and, seven days post-confluency, they are incubated with 5-FU+iri (and pre-treated with vehicle or mucin secretion inhibitors in Figure 4A) for the corresponding time points (starting for 24 hr treatment and continuing with the subsequent time points). Lysates and extracellular medium are finally collected at the same time. This procedure alleviates any issues that might arise due to time dependent release in export of mucins under baseline or stimulated secretion. We have added the protocol for these experiments in supplementary figure 1 and 4; and changed the labels of Figure 4 (New Figure 4, Figure 1 —figure supplement 1A and Figure 4 —figure supplement 1A). We have also included specific description in the Materials and methods and extended this part in the results.

“Materials and methods

Secretion assay and western blot (WB)

HT29 cells were grown to confluency and, seven days post-confluency, cells were incubated with 5-FU+iri. (25 µg/mL 5-Fu + 10 µg/mL Irinotecan in normal tissue culture medium) alone or in combination with mucin secretion inhibitor SN-6. […] Uncropped gels presented in Figure 1 and Figure 4 are provided as source data (Figure 1_source data 1, Figure 4_source data 1).”

“Results

[…] Briefly, differentiated HT29-M6 cells were treated with 5-FU+iri. (50 µg/mL 5-Fluorouracil + 20 µg/mL Irinotecan diluted in normal tissue culture medium) and MUC5AC levels monitored over time (0, 1, 3, 6 and 24 hours) by western blot (WB) (Scheme in Figure 1 —figure supplement 1A). Our results reveal that 5-FU+iri. treatment strongly increases MUC5AC intracellular levels (Figure 1A). […] MUC5AC secretion was also tested in these cells by collecting the extracellular medium at the same time points. Importantly, the first time point (time 0) represents HT29 cells not treated with 5-FU+iri. or mucin secretion inhibitors (Scheme in Figure 4 —figure supplement 1A). Thus, our results revealed that 5-FU+iri treatment caused an increase in mucin secretion (60 – 80% compared to time 0), which was blocked by SN-6 (Relative levels: 1.8 vs 1.0, respectively) (Figure 4A).”

2. There is still some confusion over the link between Kchip3 expression, mucin secretion, chemo treatment and CRC patient outcome. The authors presumably already have images showing increased and decreased mucin secretion in the cell lines where kchip3 has been mutated or overexpressed relative to controls-showing these data would greatly strengthen the paper. Additionally, if the authors could include some clarification/discussion that the patient data is based on gene expression rather than mucin secretion, and does not include information on chemotherapy treatment, this may resolve some of the confusion.

This paper is a research advance to our previous publication “KChIP3 coupled to ca^2+^ oscillations exerts a tonic brake on baseline mucin release in the colon”, where we clearly showed that modulation of KChIP3 levels affects mucin secretion. We now state in the discussion and results that the patient data is based on gene expression and does not include information on chemotherapy treatment rather than the disease-free survival parameter.

Results

“To directly assess this possibility, we evaluated whether MUC5AC expression levels discriminates patients’ outcome in a well-annotated CRC cohort (GSE14333, Jorissen cohort (18)). Importantly, although this public database does not include information on chemotherapy treatment, we could analyse the Disease-free survival (DFS) of patients depending on the gene-expression levels, a parameter that is related to tumour chemoresistance. However, DFS analysis revealed no significant differences between patients with high or low MUC5AC levels (Figure 1F), strongly suggesting that factors controlling mucin secretion, and not intracellular mucin levels, might be relevant biomarkers for CRC prognosis.”

Discussion

“[…] Our gene-expression analyses of a public CRC cohort, however, revealed that mucin levels do not reflect the prognosis of CRC patients. […] Our gene-expression data reveal that low levels of KChIP3, which negatively regulates baseline secretion (7), is an important risk factor for untreated tumours with high MUC5AC expression levels.”

Reviewer #1 (Recommendations for the authors):The authors have addressed several of the concerns raised and show nicely that the secreted material does not contain dead or detached cells. They also provide important information on mucin expression upon treatment of cells although the effect of the inhibitor SN6 alone in reducing mucin expression is not discussed. The main issue with secretion is addressed in relation to 5-FU+Iri and SN6 treatment, but not to the most central component of the study, KChiP3, relevant data is however referred to. The secreted mucins are detected by WB at a size of above 450 kDa and as the theoretical size of the naked protein in the range of 580 kDa it indicates to be the secreted product, however with limited glycosylation (a gel stained by Alcian blue would show this).

We thank the reviewer for these comments. As we have discussed above, data showing that 5-FU+iri promotes mucin secretion is included in figure 4A. We now include extended description of the experimental protocol (also added in Figure 1 —figure supplement 1A and Figure 4 —figure supplement 1A) and few additional statements in the Results section.

Materials and methods

“Secretion assay and western blot (WB)

HT29 cells were grown to confluency and, seven days post-confluency, cells were incubated with 5-FU+iri. (25 µg/mL 5-Fu + 10 µg/mL Irinotecan in normal tissue culture medium) alone or in combination with mucin secretion inhibitor SN-6. […] Uncropped gels presented in Figure 1 and Figure 4 are provided as source data (Figure 1_source data 1, Figure 4_source data 1).”

Results

“[…] Briefly, differentiated HT29-M6 cells were treated with 5-FU+iri. (50 µg/mL 5-Fluorouracil + 20 µg/mL Irinotecan diluted in normal medium) and MUC5AC levels monitored over time (0, 1, 3, 6 and 24 hours) by western blot (WB) (Scheme in Figure 1 —figure supplement 1A). Our results reveal that 5-FU+iri. treatment strongly increases MUC5AC intracellular levels (Figure 1A). […] MUC5AC secretion was also tested in these cells by collecting the extracellular medium at the same time points. Importantly, the first time point (time 0) represents HT29 cells not treated with 5-FU+iri. or mucin secretion inhibitors (Scheme in Figure 4 —figure supplement 1A). Thus, our results revealed that 5-FU+iri treatment caused an increase in mucin secretion (60 – 80% compared to time 0), which was blocked by SN-6 (Relative levels: 1.8 vs 1.0, respectively) (Figure 4A).”

However, the discrepancy in the data on mucus secretion partly remains as the results in Figure 1B, 4A and 4B show enhanced intracellular MUC5AC upon 5-FU+Iri treatment (referred to as production as it is not dependent on expression), but the data in Figure 3C with reduced number and size of vesicles is not in line. Mucus secretion was further analyzed using general staining (Alcian blue) of treated cells and organoids. The data on cells is very hard to interpret while the stained organoid shows secreted mucus. In this context treatment of the organoid with 5-FU+Iri would have been a much better representation of enhanced secretion upon treatment compared to the fixed cells.

We disagree that the data in Figure 3C (reduced number and size of granules/vesicles in cells treated with 5-FU + iri.) is not in line with the rest of the paper. As shown in our previous publication for which this paper is a research advance (Cantero-Recasens, *eLife*, 2018), the anti-MUC5AC antibody used for IF only detects the mature granules (close to the plasma membrane). In addition, it has been shown that after maximum stimulated secretion, mucin stores require 24 hr to be fully restored (Zhu et al., PLoS One, 2015). So, a decrease in the numbers (or size) of mucin granules reflects the enhanced mucin secretion by 5-FU + iri. treatment.

The link between enhanced mucin expression leading poor prognosis for CRC patients and reduced effects of treatment depending on mucus secretion is explained by differentiation, but this is a main concern in the concept and could benefit from even further discussion.

[Editors' note: further revisions were suggested prior to acceptance, as described below.]We are pleased to notify you that it is now acceptable for publication, provided that the authors include numbers of samples and replicates for all figures (as Figure 4A) and a brief explanation, as given in the response letter, why there is a variation observed in Figure 3C compared to other results presented.Reviewer #1 (Recommendations for the authors):The manuscript is now improved by text alterations. It is unfortunate that no extra data is provided proving mucus secretion to affect CRC treatment, but the text now includes most of the information needed for proper evaluation by readers. In this context I would find it important to include numbers of samples and replicates for all figures (as Figure 4A) and to include a brief explanation, as given in the response letter, why there is a variation observed in Figure 3C compared to other results presented to be fully transparent. With these additions I do find it acceptable for publication.

We thank the reviewer for the supporting comments to enable publication of our manuscript. As suggested by the reviewer, we now include the number of replicates in all figures. We have also added a sentence in the statistics section of Materials and methods, which states that “All data are means ± SEM of a minimum of three biological replicates (N ≥ 3)”. We have also provided a brief explanation of how a reduction in mucin granules (as observed in figure 3C) fits well with the other results of the paper.

“Analyses of images using FIJI software (22) revealed that cells treated with 5-FU+iri. contained significantly fewer MUC5AC-containing particles than control cells (75.7 vs. 100.7 particles/field, p=0.002) with a reduced average size (1.12 µm3 vs. 2.48 µm3) (Figure 3C, quantification in Figure 3D). It is important to note that this anti-MUC5AC antibody mainly recognizes the mature mucin granules (close to the plasma membrane, as shown in (7)), which can explain the apparent contradiction with the effect of 5-FU+iri. treatment on mucin production (Figure 1). Besides, it has been previously shown that after maximum stimulated secretion, mucin stores require 24 hours to be fully restored (23). Thus, this decrease in the numbers (and size) of mucin granules is in accordance with enhanced mucin secretion by 5-FU + iri. treatment as shown in figure 4.”